# The Identification of Metabolites and Effects of Albendazole in Alfalfa (*Medicago sativa*)

**DOI:** 10.3390/ijms21165943

**Published:** 2020-08-18

**Authors:** Lucie Raisová Stuchlíková, Martina Navrátilová, Lenka Langhansová, Kateřina Moťková, Radka Podlipná, Barbora Szotáková, Lenka Skálová

**Affiliations:** 1Department of Biochemical Sciences, Faculty of Pharmacy in Hradec Králové, Charles University, 500 05 Hradec Králové, Czech Republic; stuchli2@faf.cuni.cz (L.R.S.); navratimart@faf.cuni.cz (M.N.); szotakova@faf.cuni.cz (B.S.); 2Laboratory of Plant Biotechnologies, Institute of Experimental Botany, Czech Academy of Sciences, 165 02 Praha 6 - Lysolaje, Czech Republic; langhansova@ueb.cas.cz (L.L.); motkova@ueb.cas.cz (K.M.); podlipna@ueb.cas.cz (R.P.)

**Keywords:** anthelmintics, drug metabolism, UHPLC-MS/MS, drug phytotoxicity, drugs in the environment

## Abstract

Albendazole (ABZ), a widely used anthelmintic drug, enters the environment mainly via livestock excrements. To evaluate the environmental impact of ABZ, the knowledge of its uptake, effects and metabolism in all non-target organisms, including plants, is essential. The present study was designed to identify the metabolic pathway of ABZ and to test potential ABZ phytotoxicity in fodder plant alfalfa, with seeds and in vitro regenerants used for these purposes. Alfalfa was chosen, as it may meet manure from ABZ-treated animals in pastures and fields. Alfalfa is often used as a feed of livestock, which might already be infected with helminths. The obtained results showed that ABZ did not inhibit alfalfa seed germination and germ growth, but evoked stress and a toxic effect in alfalfa regenerants. Alfalfa regenerants were able to uptake ABZ and transform it into 21 metabolites. UHPLC-MS/MS analysis revealed three new ABZ metabolites that have not been described yet. The discovery of the parent compound ABZ together with the anthelmintically active and instable metabolites in alfalfa leaves shows that the contact of fodder plants with ABZ-containing manure might represent not only a danger for herbivorous invertebrates, but also may cause the development of ABZ resistance in helminths.

## 1. Introduction

Due to their wide usage in intensive agri- and aquaculture production, veterinary drugs represent important sources of environmental pollution. These drugs can reach the environment through treatment processes, the inappropriate disposal of used containers, unused medicine or livestock feed, as well as manufacturing processes. The application of drugs in intensively reared livestock represents the main route of their entry into an environment. When a drug (or its metabolites) is excreted by animals, it passes into the environment directly or indirectly. Direct entry occurs via the treatment of pasture-reared animals that excrete drug residues straight into the environment. The indirect route consists of the application of manure and slurry originating from treated animals to the field [1].

Once in the environment, veterinary drugs can enter and affect non-target species, including plants. Drug effects in plants involve not only biochemical and physiological disruption based on interaction with macromolecular or cellular targets, but also the disruption of signalling pathways. Modifications of gene expression have been revealed thanks to the development of transcriptomics and proteomics analyses, with certain modifications shown to be important mechanisms of plant responses to drugs and other xenobiotics [2]. While the phytotoxic effects of veterinary antibiotics have been intensively studied and reported [1], other veterinary drugs have attracted less attention. Although anthelmintics, drugs against parasitic worms, rank among the most common and frequently used veterinary drugs, not many studies have been undertaken dealing with their phytotoxicity. Wagil et al. [3] have observed no adverse effect of flubendazole and fenbendazole on the growth of duckweed (*Lemna minor*). On the other hand, significant phytotoxicity of ivermectin on *Sinapis alba* was revealed [4]. In the model plant *Arabidopsis thaliana,* exposure to ivermectin caused changes in the transcription level of numerous genes involved in the response to salt, osmotic and water deprivation stress, as well as in ion homeostasis and defence responses to pathogens [5]. Fenbendazole also significantly affected gene and protein expression in *A. thaliana* plants [6]. Neither albendazole (ABZ) nor monepantel inhibited seed germination in *Sinapis alba* [7,8], but further information about their effects in plants is lacking. The phytotoxicity of other anthelmintics has not been studied yet [1].

Like other organisms, plants defend against the potentially negative effects of drugs by transforming them into metabolites. Ideally, the metabolites are less lipophilic and less toxic than the parent compound; however, some metabolites may show higher biological activity/toxicity or instability [9]. Knowledge of the biotransformation pathway is therefore essential to evaluate the impact of drugs on plants and the environment. In our previous studies, the uptake and biotransformation of the anthelmintic drug ABZ in three plant species (*Campanula rotundifolia*, *Phragmites australis* and *Plantago lanceolata*) was tested using two plant model systems (regenerants and cell suspensions). All the tested plants were able to uptake and transform anthelmintics via various reactions, but large inter-species differences were observed [10,11,12]. Whole plant regenerants have been demonstrated as an adequate model for qualitative as well as semiquantitative evaluations [11]. Because biotransformation pathways depend not only on the structure of the drug but also on the species, drug biotransformation should be tested on all species that may meet the drug.

For these reasons and following the preceding studies, we decided to study the biotransformation of ABZ in regenerants of alfalfa (*Medicago sativa)*. In addition, the effects of ABZ on seed germination and proline accumulation in alfalfa was tested. The ABZ concentration range used (0.01–10 μM) corresponds with estimated ABZ concentrations in soil around faeces from ABZ-treated livestock. Alfalfa was chosen as a representative of common fodder plants, which might be exposed to ABZ via excrements of grazing animals or via manure used for field fertilization. Fodder plants, containing traces of ABZ and its metabolites, might by consumed by animals infected with helminths. Contact of these helminths with sub-lethal doses of anthelmintics may encourage the development of resistant strains of helminths in livestock [13,14]. This represents another possible negative impact of the presence of ABZ in plants and emphasizes the importance of studying ABZ absorption, effects and biotransformation in fodder plants.

## 2. Results and Discussion

The benzimidazole anthelmintic ABZ, one of the most used drugs in livestock farming, together with its metabolites are excreted from treated animals and enter the environment. There is little information on the stability of ABZ in faeces, but the benzimidazoles are relatively stable compounds and it is probable that they retain their structural basis for a long time [13,15]. Moreover, ABZ was detected in relatively high concentration in wastewater and both ozonation and photolysis had only little effect on its degradation [16]. Therefore, ABZ belongs to important microcontaminants and its environmental fate deserve detailed study.

### 2.1. Effect of ABZ on Germination of Alfalfa Seeds

With an aim to reveal the potential phytotoxicity of ABZ, the effects of ABZ on germination and germ growth was tested. Seeds of alfalfa and an ABZ concentration range of 0–10 µM were used. The results showed no significant toxicity of ABZ on alfalfa seeds (see Figure 1). In a previous study, ABZ did not inhibit germination of seeds of white mustard (*Sinapis alba*) [7]. It therefore seems very likely that ABZ does not exhibit a toxic effect on seeds and germ growth. Nevertheless, the effects of ABZ in mature plants have not been tested yet, and for this reason we decided to study ABZ in alfalfa regenerants.

### 2.2. Effect of ABZ on Proline Accumulation in Alfalfa Regenerants

In the plants exposed to ABZ (in various concentrations) and in control plants, the level of proline was analysed.

Proline, an essential amino acid, serves as stress indicator, as its accumulation increases in higher plants under several environmental stresses [17,18]. In alfalfa roots, lower ABZ concentrations mildly increased the proline accumulation, while higher ABZ concentrations decreased proline accumulation in the leaves (Figure 2). This indicates that lower ABZ concentrations evoked a defence response, but higher concentrations might be toxic, an effect which is manifested by the inhibition of amino acid synthesis in leaves.

Contrary to ABZ, long-term exposition of *Medicago sativa* regenerants to monepantel did not change the proline accumulation, neither in the leaves nor the roots [8]. However, ivermectin increased the proline accumulation in a *Plantago lanceolata* cell suspension [19].

### 2.3. ABZ Biotransformation in Alfalfa Regenerants

The major aim of the present study was to reveal the metabolic pathway of ABZ in alfalfa. Using UHPLC-MS/MS analysis, the ABZ metabolites were identified in alfalfa regenerants exposed to an ABZ concentration range (0.01, 0.1, 1 and 10 μM) for 6 weeks. A total of 21 metabolites were identified in the leaves and 13 metabolites in the roots. The description of all the metabolites is presented in Table 1; the proposed scheme of the ABZ metabolic pathway in alfalfa is given in Figure 3. The parent drug ABZ was detected at *m/z* 266 [M+H]^+^ (t_R_ = 7.23 min.) and with a product ion *m/z* 234 (the typical neutral loss (NL) of methanol *Δm/z* 32). S-oxidation represents the initial steps of Phase I of the ABZ biotransformation, leading to ABZ sulfoxide (M10; ABZSO; *m/z* 282 [M+H]^+^). ABZSO was consequently converted via a second S-oxidation to ABZ sulfone (M14; ABZSO_2;_ (*m/z* 298 [M+H]^+^).

MS/MS spectra contained a typical NL of Δ*m/z* 32 (methanol) and Δ*m/z* 42 (propene) for ABZSO and ABZSO_2_. Hydroxylation was the third type of Phase I reaction of the ABZ biotransformation as ABZ sulfone with a hydroxyl group at arene (*m/z* 314 [M+H]^+^) was found (M6). In two cases, when the measured data did not allow us to distinguish S-oxidation and hydroxylation, the reaction was assigned as +O (M16, M21). Hydrolysis represented the other type of Phase I reaction of the ABZ biotransformation detected in alfalfa.

In Phase II, the parent ABZ as well as the ABZ metabolites formed via Phase I underwent *N*-glycosidation, leading to the formation of M1, M5, M7, M9, M11, M20, M22, M25, M29 and M30. The NLs of Δ*m/z* 162, characteristic for hexose, Δ*m/z* 32, characteristic for methanol (with exception of hydrolyzed metabolites), and Δ*m/z* 42, characteristic for propene, were observed in the tandem mass spectra of all these conjugates.

In addition, *O*-acetylation was observed in several metabolites (M2, M4, M8, M16, M21 and M31-33). An NL of Δ*m/z* 220 corresponds to *O*-acetyl-glycoside. The formation of some metabolites required a sequence of six reactions (two step S-oxidation, hydrolysis, hydroxylation, glycosidation and acetylation) and their structure is quite complicated. Some of the metabolites are constitutional isomers, which have the same substituents but in different positions on the benzimidazole core. These metabolites have the same *m/z* ratios, but different chemical properties, which are manifested by different retention times (t_R_) in the chromatographic analyses. That is the case of the three newly detected ABZ metabolites, M31–M33.

The presence or absence of individual metabolites in the roots and leaves of alfalfa exposed to ABZ is shown in Table 2. The scheme of the ABZ metabolic pathways occurring in alfalfa is shown in Figure 3. As ABZSO and ABZSO_2_ are commercially available, the standards were purchased and used for quantification of these two main metabolites in alfalfa. Other metabolites are not available, so other metabolites could be only semi-quantified (using the area under curve and an internal standard). The concentrations of the main metabolites ABZSO and ABZSO_2_ are presented in Table 3. The most abundant metabolite found at all concentrations in both parts of the plants was ABZ sulfoxide (ABZSO, M10). ABZ S-oxidation represents the universal biotransformation of ABZ, as ABZSO was the main metabolite of ABZ in all the plant as well as animal species tested.

While ABZ metabolism is very simple in mammals, in plants ABZ metabolism is rich and extensive. Besides the two-step oxidation, several other biotransformation reactions occur in plants: hydrolysis, hydroxylation, glycosidation and acetylation. Combinations of these reactions together with their various positions in the ABZ molecules lead to the formation of a wide variety of different metabolites. Overall, 37 ABZ metabolites have been identified so far in plants. However, the number and structure of individual metabolites differ significantly among species. The lowest number of ABZ metabolites (only 10) was found in a reed (*Phragmites australis)* cell suspension [10]. Interestingly, four of these represent atypical metabolites (glucosylglucosides, xylosylglucosides) which have not been found anywhere else, and thus were not previously incorporated in the numbering of ABZ metabolites. In cells of harebell (*Campanulla rotundifolia*), 24 ABZ metabolites were identified [12], with 18 being different to the ABZ metabolites detected in reeds. In this study, the numbering of ABZ metabolites was set, and the numbers assigned here have been retained in further studies. When ABZ metabolism was studied in whole plant regenerants of plantain (*Plantago lancelolata*), 18 ABZ metabolites were found [11]. Contrary to the alfalfa regenerants, the most complicated metabolites (formed subsequently via a two-step S-oxidation, hydrolysis, hydroxylation, glucosidation and acetylation) were not detected in plantains. On the other hand, seven different metabolites were found in plantains which were not detected in alfalfa.

Overall, the results of the study of ABZ metabolism in several plant species verified the considerable richness and variety of the biotransformation reactions in plants. Significant inter-species differences in ABZ metabolism in plants were observed, which indicates large differences in genes and/or expression of xenobiotic metabolizing enzymes among plant species [10,11,12]. The finding of the parent compound ABZ together with anthelmintically active ABZSO [20,21] and several instable metabolites (all glucosides) in the leaves of alfalfa shows that ABZ’s contact with fodder plants might represent not only a danger for herbivorous invertebrates, but also a way for the development of ABZ-resistant helminths in livestock.

## 3. Materials and Methods

### 3.1. Chemicals

Albendazole (ABZ; analytical standard, ≥98%), albendazole sulphoxide (ABZSO; VETRANAL^®^, analytical standard, purity N/A), mebendazole (MBZ; internal standard, ≥98%), L-proline and other chemicals (UHPLC, MS or analytical grade) were purchased from Sigma-Aldrich (Prague, Czech Republic). Albendazole sulphone (ABZSO_2;_ analytical standard, purity N/A) was purchased from the Toronto Research Chemicals Inc. (North York, ON, Canada). Stock solutions of all benzimidazoles (10 mM) were prepared in dimethyl sulfoxide (DMSO) and stored at 4 °C in the dark.

### 3.2. Plant Material and Their Cultivation

Seeds of alfalfa (*Medicago sativa*, var. Ezzelina) purchased form AROS-osiva s.r.o. (Prague, Czech Republic) were sterilized in 70% ethanol for 1 min, followed by a 1% sodium hypochlorite solution supplemented by 0.02% detergent TWEEN 20 for a period of 10 min. The sterile seeds were washed 5-times in sterile distilled water and allowed to germinate on an agar MS medium [22,23] at 25 °C, with a 16-h photoperiod at 72 µmol of photons/m^2^/s. The germinated plants were transferred to fresh medium supplemented with ABZ and cultivated in Magenta boxes under the same conditions for 6 weeks.

### 3.3. ABZ Phytotoxicity

The ABZ effects on seed germination and roots growth was tested using the acute toxicity test modified according to [24]. In the cultivation room, the seeds of *Medicago sativa* were placed on filter paper and soaked with a dilution medium in Petri dishes (100 mm in diameter). The dilution medium was prepared in correspondence with ČSN EN ISO 6341 and ČSN EN ISO 7346-2, and contains 4 mM CaCl_2_, 1 mM MgSO_4_, 1.5 mM NaHCO_3_ and 0.15 mM KCl salts, pH 7.6–8.0. The medium was further supplemented with ABZ in four concentrations (10 µM, 1 µM, 0.1 µM or 0.01 µM), with non-supplemented medium used as the control. Twenty-five seeds and 5 mL of dilution medium was placed into one dish. Five Petri dishes, each with one of the tested ABZ concentrations, or without ABZ (solvent only; control), were seeded. The length of the seedlings was measured in order to evaluate the phytotoxicity of the ABZ after 4 days of incubation in the dark at 24 °C.

Proline content as a stress indicator was measured according to the modified method [25]. The lyophilized and homogenized roots and leaves of *Medicago sativa* (20 mg DW) grown in vitro in medium with or without ABZ (final concentrations 0.01, 0.1, 1.0 and 10 µM, pre-dissolved in DMSO) were homogenized in 40% (*v*/*v*) ethanol and incubated overnight. The extracts were centrifuged at 13,000× *g* for 10 min, and the supernatants were incubated in a water bath at 95 °C in a reaction mixture (ninhydrin 1% *w*/*v*, acetic acid 60% *v*/*v*, ethanol 20% *v*/*v*). After 20 min of incubation, the mixture was cooled and the absorbance was measured at 520 nm using a microplate reader (Tecan Infinite 200, Tecan Group Ltd., Switzerland). Each sample was measured in 8 replicates, and two independent experiments were performed. Proline calculation was based on a calibration curve made with L-Proline as the standard.

### 3.4. ABZ Uptake and Biotransformation

To evaluate the uptake and biotransformation of ABZ, the in vitro regenerants were cultivated in a medium supplemented with ABZ (final concentrations 0.01; 0.1; 1.0 and 10 µM, pre-dissolved in DMSO). The control plants were cultivated in a medium supplemented with DMSO only. The plant roots and leaves from *Medicago sativa* were collected after 6 weeks. All samples were frozen (−80 °C) before the LC/MS analysis and prepared in triplicate (*n* = 3). The plant tissues were homogenized using a FastPrep-24 homogenizer (MP Biomedicals, Santa Ana, CA, USA). The homogenates were subjected to liquid–liquid extraction (LLE) according to the method described previously [10,26]. The supernatants were evaporated to dryness using the concentrator Eppendorf plus (30 °C) (Eppendorf, Hamburg, Germany). The dried extracts were stored (−20 °C) until LC/MS analyses. Biological and chemical blank samples were prepared for all the types of procedures.

### 3.5. UHPLC-MS/MS Conditions

The dry samples were quantitatively reconstituted in the mixture water/acetonitrile (70/30, *v*/*v*) by sonication, and filtrated using syringe filters with a PTFE membrane. One microliter of the samples was injected into a UHPLC-MS system. UHPLC (Nexera; Shimadzu, Kyoto, Japan) was optimized using a Zorbax RRHD Eclipse Plus 95Å C18 column 150 × 2.1 mm, 1.8 µm (Agilent Technologies, Waldbronn, Germany), at a temperature of 40 °C, flow rate 0.4 mL/min and injection volume 1 µl. The mobile phase consisted of water (A) and acetonitrile (B), both with the addition of 0.1% formic acid (MS grade). The linear gradient was as follows: 0 min—15% B, 8 min—40% B, and 10 min—95% B followed by 1 min of isocratic elution. The QqQ mass spectrometer with an ESI ion source (LC-MS-8030 triple quadrupole mass analyser; Shimadzu, Kyoto, Japan) was used with the following setting of tuning parameters: capillary voltage 4.5 kV, heat block temperature 400 °C, DL line temperature 250 °C, the flow rate of drying gas was 12 L/min and 2.5 L/min of nebulizing gas, respectively. The mass spectrometer was operated in MRM mode by monitoring selected transitions. These transitions were obtained from previous study [12]

The detected metabolites were identified based on the presence of the protonated molecules [M+H]^+^ and the interpretation of their product ion spectra (MS^2^), as described in Appendix A. Argon was the collision gas for the MS/MS experiments.

The standards of the metabolites (with the exception of ABZSO and ABZSO_2_) were not commercially available, and they were not prepared due to the difficulties involved in their synthesis. Therefore, their amount was not quantified.

### 3.6. Quantification of ABZ, ABZSO and ABZSO_2_

The commercially available standards of ABZ, ABZSO and ABZSO_2_ were used for quantification of these metabolites in plant samples. Their linearity was evaluated by using standard calibration plots in the range 0.13265–2.653 µg/mL for ABZ, 0.149–2.813 µg/mL for ABZSO and 0.119–2.973 µg/mL for ABZSO_2_. The calibration curve was prepared using 7 calibration points. Each point contained 2.953 µg/mL of mebendazole as the internal standard. LOD and LOQ were determined by considering the signal-to-noise ratios (S/N) of 3 and 10, respectively.

The parameters such as LOD, LOQ, slope, the correlation coefficient (r^2^) for ABZ, ABSO and ABZSO_2_ are summarized in Table 4. The precision allowed to achieve an RSD < 8.2% at all individual calibration levels for ABZ,  <10.4% for ABZSO and  <10.8% for ABZSO_2._ The stock solution of ABZ, ABSO and ABZSO_2_ in DMSO were 1 mM. All calibration standard solutions were prepared in 30% ACN. All data are presented as the arithmetic mean ± SD (*n* = 3) and were acquired using LabSolution LCMS software ver. 5.93 (Shimadzu, Kyoto, Japan).

### 3.7. Statistical Analysis

The reported data are expressed as the mean ± SD (3–6 replicates). Statistical comparisons were carried out using the multiple t-test in GraphPad Prism software, version 8.4.3 (San Diego, CA, USA). The differences were considered significant at *p* < 0.05.

## 4. Conclusions

ABZ did not inhibit alfalfa seeds germination and germ growth, but evoked stress and a toxic effect in alfalfa regenerants. Alfalfa regenerants were able to uptake ABZ and transform it into 21 metabolites. UHPLC-MS/MS analysis revealed three new ABZ metabolites that have not been described so far. All ABZ metabolites (with the exception of ABZSO_2_) can be considered as anthelmintically active and/or instable. Their presence in alfalfa leaves might contribute to negative effects on herbivorous invertebrates as well as to drug resistance development in helminths. With respect to these facts, leaving of ABZ-treated animals on pasture or using manure from ABZ-treated animals to fertilize alfalfa fields can be considered risky.

## Figures and Tables

**Figure 1 ijms-21-05943-f001:**
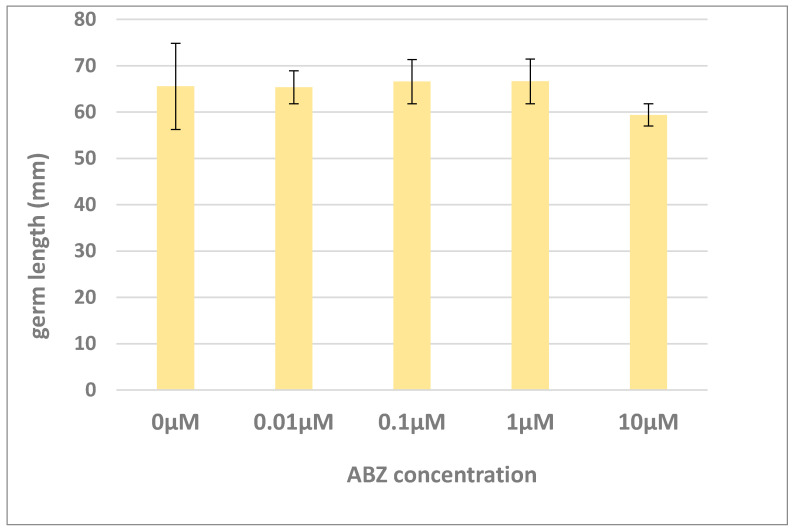
Germ length of alfalfa seeds exposed to albendazole (ABZ) in various concentrations (0–10 µM). The data represent the mean ± SD. The controls (0 µM) were exposed to solvent DMSO only.

**Figure 2 ijms-21-05943-f002:**
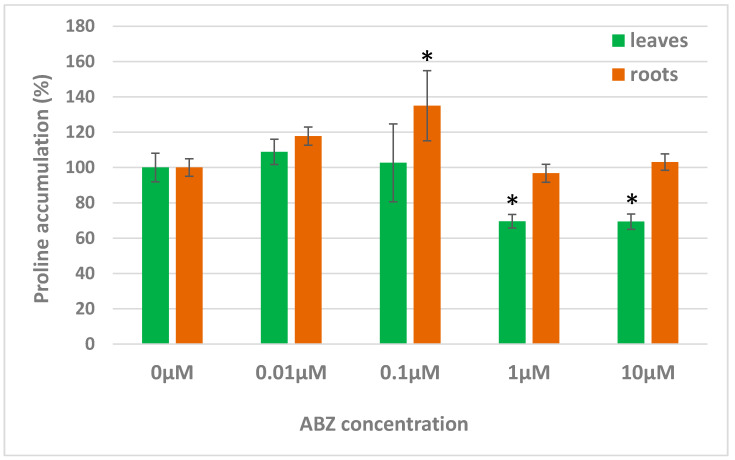
Accumulation of proline in plants exposed to ABZ in various concentrations (0–10 µM). The data represent the mean ± SD (*n* = 6) expressed as percentage of the control plants (exposed to 0 µM, solvent DMSO only). Significant changes (*p* < 0.05) are marked with an asterisk.

**Figure 3 ijms-21-05943-f003:**
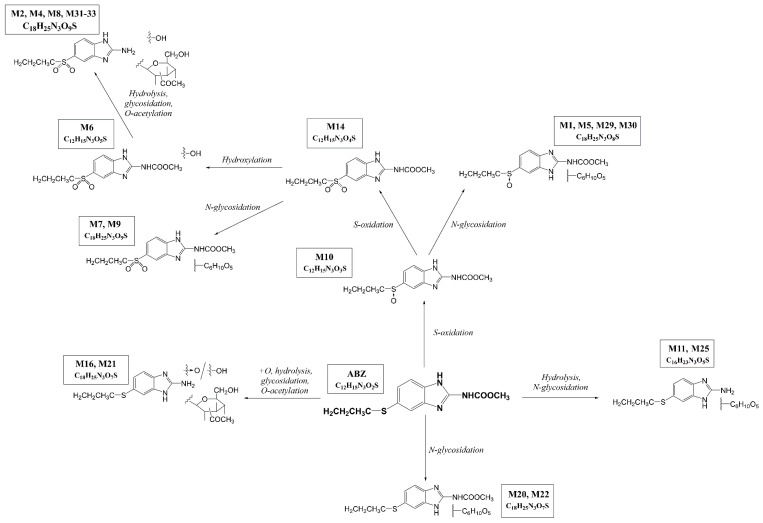
Scheme of the proposed metabolic pathway of ABZ in alfalfa.

**Table 1 ijms-21-05943-t001:** List of the main peaks for the ABZ biotransformation samples detected by UHPLC-MS/MS, with their retention times, theoretical values of [M+H]^+^ ions in the ESI positive-ion mode, molecular formula, product ions and designation of metabolites formed in the alfalfa regenerants. The newly detected ABZ metabolites are printed in bold.

t_R_ [min]	Theoretical *m/z* Values of [M+H]^+^ Ions	Molecular Formula	Description of Metabolite Formation		Product Ions of [M+H]^+^, *m/z*	Metabolite Designation
			Phase I	Phase II		
1.96	444.1435	C_18_H_25_N_3_O_8_S	S-oxidation	*N*-glucosidation	282, 240, 208, 191, 159	M1
**2.04**	**460.1384**	**C_18_H_25_N_3_O_9_S**	**2*S-oxidation, hydrolysis, hydroxylation**	**Glucosidation,** ***O*-acetylation**	**240, 198**	**M31**
2.09	460.1384	C_18_H_25_N_3_O_9_S	2*S-oxidation, hydrolysis, hydroxylation	Glucosidation, *O*-acetylation	240	M2
2.18	444.1435	C_18_H_25_N_3_O_8_S	S-oxidation	*N*-glucosidation	282, 240, 208, 191, 159	M30
2.24	444.1435	C_18_H_25_N_3_O_8_S	S-oxidation	*N*-glucosidation	282, 240, 208, 191, 159	M29
2.25	460.1384	C_18_H_25_N_3_O_9_S	2*S-oxidation, hydrolysis, hydroxylation	Glucosidation, *O*-acetylation	240, 198	M4
**2.30**	**460.1384**	**C_18_H_25_N_3_O_9_S**	**2*S-oxidation, hydrolysis, hydroxylation**	**Glucosidation,** ***O*-acetylation**	**240**	**M33**
2.44	314.0805	C_12_H_15_N_3_O_5_S	2*S-oxidation, hydroxylation	−	238, 159	M6
2.46	444.1435	C_18_H_25_N_3_O_8_S	S-oxidation	*N*-glucosidation	282, 240, 208	M5
**2.61**	**460.1384**	**C_18_H_25_N_3_O_9_S**	**2*S-oxidation, hydrolysis, hydroxylation**	**Glucosidation,** ***O*-acetylation**	**240, 198**	**M32**
2.86	460.1384	C_18_H_25_N_3_O_9_S	2*S-oxidation	*N*-glucosidation	298, 266, 224, 191, 159	M7
2.99	460.1384	C_18_H_25_N_3_O_9_S	2*S-oxidation, hydrolysis, hydroxylation	Glucosidation, *O*-acetylation	240, 133	M8
3.22	282.0907	C_12_H_15_N_3_O_3_S	S-oxidation	−	240, 208, 191,159	M10
3.36	460.1384	C_18_H_25_N_3_O_9_S	2*S-oxidation	*N*-glucosidation	298, 266, 224, 191, 159	M9
3.56	370.1431	C_16_H_23_N_3_O_5_S	Hydrolysis	*N*-glucosidation	208, 166	M11
4.50	370.1431	C_16_H_23_N_3_O_5_S	Hydrolysis	*N*-glucosidation	208, 166	M25
4.74	428.1486	C_18_H_25_N_3_O_7_S	+O, hydrolysis	Glucosidation, *O*-acetylation	208	M16
5.00	298.0856	C_12_H_15_N_3_O_4_S	2*S-oxidation	−	266, 224, 159	M14
5.54	428.1486	C_18_H_25_N_3_O_7_S	−	*N*-glucosidation	266, 234	M20
6.06	428.1486	C_18_H_25_N_3_O_7_S	+O, hydrolysis,	Glucosidation, *O*-acetylation	208, 250	M21
6.56	428.1486	C_18_H_25_N_3_O_7_S	−	*N*-glucosidation	266, 234	M22
7.23	266.0958	C_12_H_15_N_3_O_2_S	−	−	234	ABZ

**Table 2 ijms-21-05943-t002:** Presence (+) or absence (−) of individual ABZ metabolites in roots and leaves of alfalfa.

	*M. sativa* Roots					*M. sativa* Leaves			
	ABZ (µM)					ABZ (µM)			
	0.01	0.1	1.0	10		0.01	0.1	1.0	10
M1	−	−	+	+			+	+	+
M30	−	−	−	+		−	−	−	−
M29	−	−	−	+		−	−	−	+
M31	−	−	−	+		−	−	−	+
M4	−	−	−	+		−	−	−	+
M32	−	−	−	+		−	−	−	−
M7	−	−	−	+		−	−	−	+
M9	−	−	−	+		−	−	−	+
M25	−	−	−	+		−	−	−	−
M21	−	−	−	+		−	−	−	−
M5	−	−	−	+		−	−	−	−
M2	−	−	−	+		−	−	−	+
M33	−	−	−	+		−	−	−	−
M8	−	+	+	+		−	−	−	+
M11	−	−	−	+		−	+	+	+
M16	−	−	−	+		−	−	−	−
M20	−	−	−	+		−	−	−	−
M22	−	−	−	+		−	−	−	−
M10	+	+	+	+		+	+	+	+
M14	+	+	+	+		+	+	+	+
M6	−	−	−	+		−	−	−	+

**Table 3 ijms-21-05943-t003:** The amount (µg of chemicals per g of dry plant tissue) of ABZ and its two main metabolites ABZSO and ABZSO_2_ in roots and leaves of alfalfa exposed to ABZ in concentrations of 0.01, 0.1, 1.0 and 10 µM for 42 days.

Exposition of Alfalfa to ABZ (µM)					
		0.01	0.1	1.0	10
**Roots**	**ABZ**	0.008 ± 0.002	0.019 ± 0.01	0.015 ± 0.005	0.162 ± 0.011
	**ABZSO**	1.9 ± 1.2	4.1 ± 1.1	113.6 ± 23.7	1073 ± 209
	**ABZSO_2_**	0.25 ± 0.12	0.54 ± 0.08	6.59 ± 1.37	42.7 ± 6.21
**Leaves**	**ABZ**	0.005 ± 0.002	0.005 ± 0.003	0.012 ± 0.005	0.006 ± 0.001
	**ABZSO**	0.444 ± 0.134	0.44 ± 0.19	3.25 ± 0.25	61.6 ± 2.19
	**ABZSO_2_**	0.094 ± 0.016	0.09 ± 0.02	0.64 ± 0.24	6.21 ± 0.10

**Table 4 ijms-21-05943-t004:** Analytical performance data of ABZ, ABZSO and ABZSO_2_ for linearity of the calibration curves, correlation coefficients and limits of detection and quantification.

	Standard Curve (y = ax + b)	r^2^	LOD (ng/mL)	LOQ (ng/mL)
ABZ	Y = (16.9401)X + (0.892919)	0.9959	0.0008	0.0028
ABZSO	Y = (0.731182)X + (0.0388037)	0.9972	0.0104	0.0350
ABZSO_2_	Y = (0.463911)X + (−0.00401502)	0.9934	0.0110	0.0367

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
