# Peer review of "The Identification of Metabolites and Effects of Albendazole in Alfalfa (Medicago sativa)"

_ijms, 2020, doi:10.3390/ijms21165943_

Round 1

Reviewer 1 Report

The study is good and very useful, thank you for doing it
But there are some notes and inquiries, thank you for answering them

Introduction:

  • Add a paragraph and reference studies on the changes that happen to the ABZ in the environment after the dung is released from the animals in the pastures. As well as the changes, that occur on the ABZ in the fermented manure, which is used to fertilize alfalfa fields.

Results and discussion:

  • Line 89 Ref. (7) it is not enough because it is about ABZ concentrations in lamb dung. The best is to support the result with other references on the use and effect of ABZ on seed germination in the laboratory.
  • Results need support with more references on the topics discussed (if applicable).
  • In the results of paragraphs 2.2 and 2.3, it is worth noting that alfalfa is presented to animals at the beginning of flowering stage, where they have the most beneficial effect for the animal. Here it was necessary to analyze the presence of ABZ and the rest of the components at this stage in the life of the plant. In this study, the analysis was done at the stage of six weeks.

Material and methods:

  • Were the seeds washed with sterile water after being sterilized with the mentioned substances? It should be mentioned
  • Line 192 There is no need to mention the source of purchasing alfalfa, because this was mentioned previously, it is sufficient only to write alfalfa seeds
  • Line 196 and 214 on which basis were concentrations chosen from ABZ? Is it compatible with the concentrations of ABZ that come with animal droppings into the environment (alfalfa fields)?
  • Line 197 Was the seed planted in a petri dish in an isolation room? It should be mentioned
  • Conclusions :
  • Line 256: This recommendation is supposed to be confirmed, for two reasons:
  • Confirm the harmful proportions of ABZ in pasture animal dung according to the area designated for grazing. Note that the manurefrom the animals is spread over the entire area of the field and is subject to environmental factors.
  • For animal dung that is indoors (in the barn), the manure is not transported directly to the field, but is stored and fermented for several months before using it in the field. During fermentation, changes occur to the composition of the manure and its components that may affect ABZ and its ratio.
  • Note: I suggest that the authors continue to work on alfalfa, but in a field experiment in which manure from animals treated with Albendazole is used, and to study the content of Albendazole in alfalfa during several stages of plant life, especially in the beginning of flowering. And to repeat the experiment two years because alfalfa is perennial. And in several cutts during each season. The results are thus more realistic and practical.

Author Response

Dear Reviewer 1,

We are grateful for the revision of our MS, thank you very much for it. We made an effort to improve our MS according your suggestions. All changes are highlighted in revised version of our MS. The comments and answers are below the individual queries and/or recommendation:

Introduction:

  • Add a paragraph and reference studies on the changes that happen to the ABZ in the environment after the dung is released from the animals in the pastures. As well as the changes, that occur on the ABZ in the fermented manure, which is used to fertilize alfalfa fields.

Unfortunately, there is no information about the decomposition of ABZ in faeces, but the benzimidazoles are relatively stable compounds and its probable that they retain their structural basis for a long time [Ref. 12, 14]. Moreover, ABZ was detected in relatively high concentration in wastewater and both ozonation and photolysis had only little effect on its degradation [Ref. 15]. This information was added at the beginning of Results and Discussion section

Results and discussion:

  • Line 89 Ref. (7) it is not enough because it is about ABZ concentrations in lamb dung. The best is to support the result with other references on the use and effect of ABZ on seed germination in the laboratory.

Ref 7 is not only about ABZ concentration in lamb dung, but it also describes no effect of ABZ on the germination of Sinapsis alba seeds.

  • Results need support with more references on the topics discussed (if applicable).

The references about ABZ stability and wastewater occurrence were added.

  • In the results of paragraphs 2.2 and 2.3, it is worth noting that alfalfa is presented to animals at the beginning of flowering stage, where they have the most beneficial effect for the animal. Here it was necessary to analyze the presence of ABZ and the rest of the components at this stage in the life of the plant. In this study, the analysis was done at the stage of six weeks.

Fields of alfalfa are usually mown at the beginning of flowering, but alfalfa in the pastures is consumed continuously by livestock. However, our experiments were performed in vitro to eliminate other factors affecting plant physiology and ABZ metabolism. Without hormonal supplementation, in vitro regenerants do not bloom.

Material and methods:

  • Were the seeds washed with sterile water after being sterilized with the mentioned substances? It should be mentioned

Yes, the sterile seeds were washed 5-times in sterile distilled water before germination. This information was added in Material and methods section.

  • Line 192 There is no need to mention the source of purchasing alfalfa, because this was mentioned previously, it is sufficient only to write alfalfa seeds

It was corrected.

  • Line 196 and 214 on which basis were concentrations chosen from ABZ? Is it compatible with the concentrations of ABZ that come with animal droppings into the environment (alfalfa fields)?

In our experiments, alfalfa plants were exposed to 0.01-10 μM i.e.0.13-132 µg of ABZ in 50ml of medium. Maximal concentration of ABZ in lamb dung after recommended dose of ABZ was 7.7 µg/g. (ref. 7). Our maximal concentration simulates situation where one plant is covered with approx. 20g of dung with maximal ABZ concentration. Lower concentrations simulate situations where the plants are exposed to less amount of dung or dung with less amount of ABZ. Overall, concentration range used (0.01-10 μM) corresponds with estimated ABZ concentrations in soil around faeces from ABZ-treated livestock. This information was added in Introduction section.

  • Line 197 Was the seed planted in a petri dish in an isolation room? It should be mentioned

The test was performed in cultivation room at control conditions. This information was added in Material and methods section.

  • Conclusions :
  • Line 256: This recommendation is supposed to be confirmed, for two reasons:

Confirm the harmful proportions of ABZ in pasture animal dung according to the area designated for grazing. Note that the manure from the animals is spread over the entire area of the field and is subject to environmental factors.

For animal dung that is indoors (in the barn), the manure is not transported directly to the field, but is stored and fermented for several months before using it in the field. During fermentation, changes occur to the composition of the manure and its components that may affect ABZ and its ratio.

We agree with reviewer that ABZ in manure probably undergoes some decomposition during storage and fermentation. However, there is no information about changes in ABZ concentration during manure storage and about the needed storage period for total decomposition. From this point of view, we consider the fertilization with manure from ABZ-treated animals as potentially risky. The last sentence of Conclusions was reformulated.

  • Note: I suggest that the authors continue to work on alfalfa, but in a field experiment in which manure from animals treated with Albendazole is used, and to study the content of Albendazole in alfalfa during several stages of plant life, especially in the beginning of flowering. And to repeat the experiment two years because alfalfa is perennial. And in several cutts during each season. The results are thus more realistic and practical.

We agree with reviewer. Nowadays, study of ABZ circulation in real farm condition is performed.

Reviewer 2 Report

The manuscript “The identification of metabolites and effects of albendazole in alfalfa (Medicago sativa)” deals with the biotransformation of albedazole (ABZ) in regenerants of alfalfa (Medicago sativa) and the potential ABZ phytotoxicity. This work is interesting because, as claimed by the authors, the biotransformation pathways depend not only by the drug, but also on the species. The authors conducted this work, more or less, as described in their previous papers on three other plant species.

Even if the topic is interesting, I have some doubts about how they conducted the analysis and reported the data in the text.  

Major revisions:

Subparagraph 2.1 – the authors did not show the data of the phytotoxicity of ABZ on germination of alfalfa seeds because the results are not statistically significant. So, this subparagraph does not add any further information! I suggest to completely remove it.

Subparagraph 2.3 – the authors claim in the abstract and conclusions that four new metabolites were reported for the first time in the metabolic pathway of ABZ. Which ones? Adding them in the results section with some explanation is relevant.    

Figure 2 – highlight the new metabolites and correct M10 formula, it is reported with the wrong number of carbons (C13H15N3O3S instead of C12H15N3O3S). Furthermore, the quality of this figure should be improved.

In my opinion, the authors should add in supplementary material the MS/MS spectra of metabolites reported in table 1.

Concerning the quantification of ABZSO and ABZSO2, in the results section (lines 145—146) authors say that the concentrations of these metabolites in roots and leaves were obtained using commercially available standards, but in materials and methods section (lines 240-244) they say that they did not use standards to quantify metabolites. It is not clear what they mean with “metabolites were semi-quantified”. Do you quantify or not these metabolites? To quantify metabolites, they need commercial standards because the quantification must be done by linear regression of peak areas of known concentrations (at least 4-5) of commercially available standards. The response of target compounds must be normalized to the response of an internal standard. Please add all information about the quantification. Furthermore, the significance analysis is missing in table 3 and it is important to conclude that differences in the amount of metabolites are statistically significant.

Conclusions- The authors say in subparagraph 2.1 that ABZ phytotoxicity results on seeds germination are not statistically significant so why they conclude that ABZ did not inhibit alfalfa seeds germination if they have no piece of evidence?

Minor revisions:

Line 59- please add references

Figure 1 – the authors should define the annotation (*) in the figure caption

Line 171 add references concerning the anthelmintic activity of ABZSO

Subparagraph 3.5 – Please correct symbols (see line 228 and 238).

Line 237 – Supplementary material???

The references should be reported according to the journal requirements.

Author Response

Dear Reviewer 2,

We are grateful for the revision of our MS, thank you very much for it. We made an effort to improve our MS according your suggestions. All changes are highlighted in revised version of our MS. The comments and answers are below the individual queries and/or recommendation:

The manuscript “The identification of metabolites and effects of albendazole in alfalfa (Medicago sativa)” deals with the biotransformation of albedazole (ABZ) in regenerants of alfalfa (Medicago sativa) and the potential ABZ phytotoxicity. This work is interesting because, as claimed by the authors, the biotransformation pathways depend not only by the drug, but also on the species. The authors conducted this work, more or less, as described in their previous papers on three other plant species.

Even if the topic is interesting, I have some doubts about how they conducted the analysis and reported the data in the text.  

Major revisions:

Subparagraph 2.1 – the authors did not show the data of the phytotoxicity of ABZ on germination of alfalfa seeds because the results are not statistically significant. So, this subparagraph does not add any further information! I suggest to completely remove it.

The results (data) were significant but the changes were insignificant. The misleading sentence was removed.

Subparagraph 2.3 – the authors claim in the abstract and conclusions that four new metabolites were reported for the first time in the metabolic pathway of ABZ. Which ones? Adding them in the results section with some explanation is relevant.    

Some of the metabolites are constitutional isomers which have the same substituents but in different positions at the benzimidazole core. These metabolites have the same m/z ratios, but different chemical properties, which are shown by different retention times (tR) in the chromatographic analyses. That is the case of three newly detected ABZ metabolites M31-M33. This information was added in the MS and these new metabolites were highlighted in Table 1 by printing in bold. The fourth metabolite formerly considered as new has been previously detected in other species. This was corrected in the MS.

Figure 2 – highlight the new metabolites and correct M10 formula, it is reported with the wrong number of carbons (C13H15N3O3S instead of C12H15N3O3S). Furthermore, the quality of this figure should be improved.

The formula was corrected. The quality of figure was improved to 600 dpi in TIFF format.

In my opinion, the authors should add in supplementary material the MS/MS spectra of metabolites reported in table 1.

Supplementary material was prepared as recommended.

Concerning the quantification of ABZSO and ABZSO2, in the results section (lines 145—146) authors say that the concentrations of these metabolites in roots and leaves were obtained using commercially available standards, but in materials and methods section (lines 240-244) they say that they did not use standards to quantify metabolites. It is not clear what they mean with “metabolites were semi-quantified”. Do you quantify or not these metabolites? To quantify metabolites, they need commercial standards because the quantification must be done by linear regression of peak areas of known concentrations (at least 4-5) of commercially available standards. The response of target compounds must be normalized to the response of an internal standard. Please add all information about the quantification. Furthermore, the significance analysis is missing in table 3 and it is important to conclude that differences in the amount of metabolites are statistically significant.

As ABZSO and ABZSO2 are commercially available, the standards were purchased and used for quantification of these metabolites in alfalfa. Other metabolites are not available, so other metabolites could be only semi-quantified (using the area under curve and internal standard). This explanation was added in Results and Discussion section and the description of procedure was corrected in Material and Methods section

Conclusions- The authors say in subparagraph 2.1 that ABZ phytotoxicity results on seeds germination are not statistically significant so why they conclude that ABZ did not inhibit alfalfa seeds germination if they have no piece of evidence?

The results significantly proved no ABZ toxicity on seed germination

Minor revisions:

Line 59- please add references

References were added

Figure 1 – the authors should define the annotation (*) in the figure caption

It was corrected

Line 171 add references concerning the anthelmintic activity of ABZSO

  • Two references of papers describing the anthelmintic activity of ABZSO were added

Subparagraph 3.5 – Please correct symbols (see line 228 and 238).

            The symbols are corrected.

Line 237 – Supplementary material???

            The Supplementary material with MS/MS spectra was completed

The references should be reported according to the journal requirements.

            The reference style was corrected.

Round 2

Reviewer 2 Report

In this revised version the authors have improved the manuscript, but I still have some comments on this new version of the manuscript.

In my opinion, the data of the effects of ABZ on germination should be showed in a graph or table, even if there is no significant toxicity. The graphical representation of the data definitely avoids possible misunderstandings.

The authors say “these metabolites have the same m/z ratios, but different chemical properties, which are shown by different retention times (tR) in the chromatographic analyses” so I think is better adding in the text that these metabolites were PUTATIVELY identified because the only evidence, that might suggest possible isomers, is the different retention time.

Concerning the quantification of the commercially available metabolites (i.e. ABZ, ABZSO and ABZSO2), this part is poorly written.  I suggest adding a new subparagraph in which authors can add details which are essential in the Internal Standard Calibration Method:

  • Certified standards suppliers
  • the substance used as Internal standard and its concentration
  • Solvent used to prepare the solutions
  • How many calibration solutions were used and their concentrations
  • Performance data of the quantification method: determination coefficient (R2), slope, intercept, LOD and LOQ

As reported in my previous revision, I still think that the authors should add the significance analysis to the quantification data reported in Table 3.

Where is the results of the “semiquantified metabolites”? The data obtained in this way make not possible further deduction and interpretation because they are not a real quantification. So i don’t understand why the authors do that, in particular because the data are not showed.  

The journal names should be reported in the reference list in abbreviated form.

Author Response

Dear Reviewer 2,

thank you very much for the detailed revision of our MS. We made an effort to improve our MS according your suggestions. All changes are highlighted in revised version of our MS. The comments and answers are below the individual queries and/or recommendation:

In my opinion, the data of the effects of ABZ on germination should be showed in a graph or table, even if there is no significant toxicity. The graphical representation of the data definitely avoids possible misunderstandings.

The Figure 1 presenting the results of seeds germination was prepared and added in the MS

The authors say “these metabolites have the same m/z ratios, but different chemical properties, which are shown by different retention times (tR) in the chromatographic analyses” so I think is better adding in the text that these metabolites were PUTATIVELY identified because the only evidence, that might suggest possible isomers, is the different retention time.

If the compounds have the identical m/z ratios, MRM transfers and fragmentation ions (verified using Q-TOF high-resolution MS analysis), but different retention times, we are sure, that these compounds are isomers.

Concerning the quantification of the commercially available metabolites (i.e. ABZ, ABZSO and ABZSO2), this part is poorly written.  I suggest adding a new subparagraph in which authors can add details which are essential in the Internal Standard Calibration Method:

  • Certified standards suppliers
  • the substance used as Internal standard and its concentration
  • Solvent used to prepare the solutions
  • How many calibration solutions were used and their concentrations
  • Performance data of the quantification method: determination coefficient (R2), slope, intercept, LOD and LOQ

The subparagraph and Table 4 with required information was added in Material and methods section.

As reported in my previous revision, I still think that the authors should add the significance analysis to the quantification data reported in Table 3.

Table 3 presents the amounts of ABZ, ABZSO and ABZSO2 in leaves and roots of plants exposed to ABZ in various concentrations. The data demonstrate the concentration-dependency of ABZ uptake and ABZSO and ABZSO2 formation. No statistical evaluation was performed as no comparison was made (in control plants, no ABZ and its metabolites were detected as supposed).

Where is the results of the “semiquantified metabolites”? The data obtained in this way make not possible further deduction and interpretation because they are not a real quantification. So i don’t understand why the authors do that, in particular because the data are not showed.  

As Table 2 demonstrates only presence/absence of the individual metabolites, the information about the semi-quantification procedure was withdrawn.

The journal names should be reported in the reference list in abbreviated form.

The form of journal names was corrected.